# Study on the Regularity of Ammonia-Related Refrigeration Accidents in China from 2010 to 2020

**DOI:** 10.3390/ijerph19148230

**Published:** 2022-07-06

**Authors:** Cong Luo, Yunsheng Zhao, Ke Xu

**Affiliations:** 1Faculty of Engineering, China University of Geosciences, Wuhan 430074, China; 2Department of Emergency Management of Hubei Province, Wuhan 430064, China; ke_xuhb@163.com

**Keywords:** ammonia-related refrigeration, accident statistics, accident prevention

## Abstract

The frequent occurrence of ammonia-related refrigeration accidents (ArRAs) restricts the safety and sustainable development of cold storage. As an essential tool for safety management, accident statistical analysis can provide a crucial decision-making basis for accident prevention and control. The present study combined descriptive statistics and comparative analysis methods to explore the characteristics and regularities of 82 ArRAs in China from 2010 to 2020. The results showed that the annual evolution of ArRAs presents a bimodal “M” mode in which 2013 and 2016 were the peaking years of accidents. The monthly distribution has an agglomeration effect, and the period from June to September had a high incidence period of accidents. The ArRAs mainly occurred in East China and Central China in the spatial dimension. Zhejiang, Shandong, Hubei, and Sichuan are the pivotal provinces for preventing and controlling ArRAs. Human factors and equipment failure are the leading causes of ArRAs. Accident numbers and casualties have inconsistent trends due to the uncertainty and variability of ArRAs’ consequences. The safety situation of ammonia-related refrigeration enterprises has improved but still needs to strive to prevent and control major accidents. This study draws valuable references for safety decision-making by ammonia-related refrigeration enterprises and safety regulators.

## 1. Introduction

With the steady development of domestic cold chain logistics and the effective promotion of the cold storage construction process, the number and scale of cold storage have increased year by year [1]. As of 2019, China has 1832 cold storage with a total capacity of 60.53 million tons, ranking third globally. China Business Industry Research Institute predicts that the total volume of cold storage in China will reach 73.22 million tons in 2021. Regarding the refrigerants used in cold storage, the proportion of cold storage using liquid ammonia refrigeration is 69.4% [2,3]. The critical issue of ammonia refrigeration cold storage is safety [4,5]. Ammonia refrigeration systems often constitute a major source of danger, and are highly susceptible to severe accidents because of ammonia’s particular physical and chemical properties, the complexity of the ammonia system equipment and facilities, and the uncertainty of personnel operation [6,7]. In 2013, the “6.3” fire and explosion accident at Baoyuanfeng Poultry Industry Co., Ltd. in Changchun, Jilin Province, China, caused 121 deaths, 76 injuries, and direct economic losses of 182 million yuan. The “8.31” major ammonia leak accident at Shanghai Weng Refrigeration Industry Co., Ltd. (Shanghai, China) caused 15 deaths, 7 serious injuries, and 18 minor injuries, resulting in direct economic losses of about 25.1 million yuan [8,9]. The rapid growth of ammonia refrigerated cold storage scale and the frequency of serious accidents have put forward higher requirements for the safety management of ammonia-related refrigeration enterprises in China.

Accident statistical analysis is an essential tool for safety management [10,11]. Based on numerous accident cases, mathematical statistics methods are applied to explore the causes and patterns of accidents, providing a scientific basis for macro accident prediction and safety management [12,13,14]. Accident statistics analysis is widely used in hazardous chemicals, transportation, and coal mine [15,16,17,18]. Jie Hou et al. [19] analyzed 5207 hazardous chemical spill accidents in China from 2009 to 2018, and the results show a regional imbalance in hazardous chemical spill accidents in China, with most accidents occurring in coastal areas. Keping Zhou et al. [20] selected seven accident characteristics to analyze hazardous chemical accidents and also concluded that the occurrence of accidents has obvious time-domain and regional features. He-Da Zhang and Xiao-Ping Zheng [21] investigated the statistical characteristics of 1632 hazardous chemical accidents in China from 2006 to 2010, focusing on fixed facilities and transportation types. They concluded that the equipment life cycle theory could explain the occurrence of accidents. Guizhen He et al. [22] analyzed 976 major hazardous chemical accidents recorded in China over the past 40 years to identify areas for improvement in developing integrated risk management in China. The above studies confirmed the feasibility of accident statistics and inspired the present study for statistical analysis of ammonia-related accidents (ArRAs).

Scholars have studied ammonia-related refrigeration accidents from accident causes analysis [23,24,25,26], numerical simulation of accident consequences [27,28,29], and leakage dispersion tests [30,31], which have provided valuable experience in preventing and controlling ArRAs. However, relatively few studies focus on the macroscopic statistical analysis of ArRAs. Du et al. [32] analyzed ammonia leakage accidents from 2007 to 2015 and combined the information diffusion model to calculate the annual probability of ammonia leakage. Liu Hui et al. [33] conducted a statistical analysis of liquid ammonia accidents in workplaces from 2011 to 2015 and concluded that liquid ammonia leakage accidents mainly occurred in the food, chemical, and metallurgical industries. The above analysis shows that accident statistical analysis mainly focuses on the major categories of hazardous chemicals [34]. Hence, there are few statistical analyses for ammonia accidents, especially for ArRAs from 2010 to 2020. To strengthen the statistical analysis of ammonia-related refrigeration accidents and accident pattern research have important theoretical significance and practical value in promoting ammonia-related refrigeration enterprises’ safety and sustainable development. This is the motivation for this study.

Therefore, this paper provides a statistical analysis of ArRAs in China from 2010 to 2020, aiming to dig into the spatial and temporal characteristics and patterns of accidents and point out the emphasis on accident prevention and control to provide a reference for safety macro-decision-making in the field of ammonia-related refrigeration.

## 2. Materials and Methods

### 2.1. Data Sources

Figure 1 depicts the data source, data processing, and data analysis. Accident information was mainly collected from four websites: China Chemical Safety Association (CCSA) (http://www.chemicalsafety.org.cn/), the China Chemical Safety Network (NRCC) (http://service.nrcc.com.cn/), the Safety Management Network (https://www.safehoo.com/), and the Ministry of Emergency Management (MEM) of the People’s Republic of China (PRC) (https://www.mem.gov.cn/).The Ministry of Emergency Management of the People’s Republic of China (PRC) is a new government agency established in 2018, whose official website is a common source of accident data. The “Accident and Disaster Investigation” of the MEM module can collect complete investigation reports of various accidents. China Chemical Safety Association, established on 18 May 2006, is a vital social force in promoting and developing the cause of chemical safety production in China. The Accident Alert and Accident Case of CCSA can provide detailed accident information. China Chemical Safety Network (NRCC) was established and operated by Qingdao Nuocheng Chemical Safety Technology Co., Ltd. (Qingdao, China). The Chemical Accident Statistics and Analysis of NRCC can provide abundant accident data. Safety Management Network is a large-scale portal website mainly focusing on publicizing safety work, entirely operated by Beijing Dongfang Chuangxiang Technology Co., Ltd. (Beijing, China). The “Accident Cases” module of the website has a wealth of accident information as a supplementary data source for this study.

Three screening criteria were used for accident case collection:Accident/incident occurred in ammonia-related refrigeration enterprises and ports of China;The hazard of the accident is ammonia, ammonia gas, or its flammable mixture;The accident happened between 1 January 2010 and 31 July 2020.

Based on the above criteria and search engines, the present research has collected 82 ammonia refrigeration-related accidents. A small number of incidents may result in losses due to some delayed disclosure cases and perceived causes. The collected information contains the enterprise name, the accident time, accident location, accident type, accident causes, the number of casualties, the economic loss, and the emergency measures.

### 2.2. Analysis Methods

Descriptive statistical methods and comparative analysis methods were used in this paper. Firstly, the extracted information (Enterprise name, accident time, accident location, accident type, accident cause, and the number of casualties) was transformed into an Excel-format database. Then, two types of variables were identified: categorical and numerical. The categorical variables refer to the year of occurrence, month of occurrence, province of occurrence, geographical division, and economic division, while the numerical variables refer to the number of accidents, fatalities, and injuries. Descriptive statistical and comparative methods were proposed to analyze the accident characteristics of ammonia-related refrigeration. The research idea is shown in Figure 1.

## 3. Results

### 3.1. Temporal Evolution Characteristics of ArRAs

#### 3.1.1. Annual Evolution of ArRAs

There were 82 ammonia refrigeration-related accidents in China from 2010 to 2020, resulting in 189 deaths and 1081 injuries (including poisoning). Table 1 shows the details.

Figure 2 shows that the overall annual evolution of ArRAs exhibits an “M” pattern with a double-increasing. The evolutionary stage can be divided into three phases: 2010–2013, 2014–2015, and 2016–2020. Before 2013, the number of ArRAs increased year by year and reached a peak in 2013. Between 2014 and 2015, the number of ArRAs decreased significantly, while there was a single-point upward trend in 2016. Fortunately, the number of ArRAs decreased yearly in the recent four years.

It is worth noting that 2013 accounts for the largest number of accident fatalities at 80.95% of the total. The reason is that several high-impact accidents caused heavy casualties in 2013, especially the Jilin Baoyuanfeng “6.3” Major Fire Accident and Shanghai “8.31” Major Liquid Ammonia Leakage Accident. Some representative accident details are shown in Table 2. However, around 2013, the distribution of ArRAs fatalities is relatively flat.

In terms of the annual evolution of injuries, the highest number of injuries is in 2012, with more than 500, followed by 2010 and 2013, with 190 and 137 respectively. Furthermore, the injuries of ArRAs from 2010 to 2013 accounted for 78.63% of the total injuries in the past decade. The higher number of injuries in 2012 was mainly due to an ammonia leak in Hubei that resulted in multiple poisonings, but due to emergency rescue timely, there were no fatalities in this accident, as shown in Table 2. Before 2016, the variation trend of accident numbers and casualties indicates apparent synchronization. In other words, the number of casualties increases with the growth of accident numbers and decreases with the decreased accident numbers. However, since 2017, the trend has not fully coincided. It can be seen that after 2017, the number of accidents decreased year by year, but the number of accident deaths and injuries has shown a weak upward trend.

#### 3.1.2. Monthly Evolution of ArRAs

The monthly distribution of the number of accidents, fatalities, and injuries is shown in Table 3. ArRAs mainly occurred in the second and third quarters, with 57 accidents, accounting for 69.51%. The highest number of accidents occurred in July, with a cumulative total of 12, followed by April, June, August, and September, with a cumulative total of 10 accidents in a single month, as shown in Figure 3. In contrast, February had the lowest number of accidents, only one.

June had the highest accident fatalities, with 65.08% of total fatalities, whose injuries also accounted for 29.79%. It is followed by August and November, with a relatively high number of deaths, amounting to 10.58% and 11.11%. Meanwhile, October had the highest number of accident injuries, accounting for 44.40%. In comparison, December and January had a relatively low number of fatalities and injuries. It is evident that the variation trend of the number of accidents, accident fatalities, and injuries is not synchronized. As we can see, the accident number reached its highest point in July, but the number of fatalities and injuries was low. Instead, the number of fatalities and injuries peaked in June and October. However, the evolution trend of the three variables synchronized merely in January and February. January and February had fewer accidents than other months, and their accident casualties were significantly lower than in other months. Separately, the trend of accident fatalities and injuries synchronized monthly, which means that changes in the number of accident injuries were consistent with changes in fatalities.

### 3.2. Spatial Distribution Characteristics of ArRAs

#### 3.2.1. Distribution in 31 Administrative Districts

ArRAs in 2010–2020 covers the majority of provinces in China, the number of which is shown in Figure 4. ArRAs mainly occurs in the eastern coastal and central inland provinces. Hubei Province and Zhejiang Province have the highest number of accidents, with nine accidents each. Shandong Province ranked second, with seven accidents. Sichuan Province has six accidents, ranking third. In addition, five accidents occurred in Liaoning Province and Henan Province, respectively. However, no accidents occurred in Qinghai, Guizhou, Shanxi, Tibet Autonomous Region, and other provinces.

The statistical distribution of the number of deaths and injuries in 31 statistical units (Hong Kong, Macao, and Taiwan are excluded in the statistics) is shown in Figure 5. Overall, casualties of ArRAs are mainly concentrated in the Northeast. Jilin Province had the most significant number of deaths, with 121. Shandong Province followed with 17 deaths. Shanghai ranks third with a cumulative death toll of 15. Hubei, Liaoning, Jilin, and Shandong provinces have the most injured people.

The cumulative number of injured people in Hubei Province was 629. Followed by Liaoning Province, the cumulative number of injured reached 134. Jilin Province has 76 people, ranking third. In addition, the cumulative number of injured people in Shandong Province also reached 67. Further comparative analysis shows that the number of accidents in Jilin Province is relatively tiny, but the fatalities and injuries are rather large. Conversely, Zhejiang province has the most accidents and has far fewer fatalities and injuries than Jilin province.

#### 3.2.2. Distribution in Seven Geographic Regions

In order to further understand the spatial distribution of ArRAs from a macro perspective, the number of accidents, fatalities, and injuries were statistically analyzed by the seven major administrative geographical regions of China: North China (NCHN), Northeast China (NECHN), East China (ECHN), Central China (CCHN), South China (SCHN), Southwest China (SWCHN), and Northwest China (NWCHN), as shown in Table 4 and Figure 6.

ArRAs mainly occurred in East China, Central China, Northeast China, and Southwest China, accounting for about 80% of the total accidents. As shown in Figure 6, East China has the most significant number of accidents, with 31 accidents, accounting for 37.80%. The number of accidents in Central China is second, accounting for 19.51%. In addition, there are few accidents in South China and Northwest China.

The accident fatalities are mainly amassed in Northeast China and East China, with the two regions accounting for about 87% of the total fatalities. Northeast China has the highest fatalities, with 121 people, accounting for 64.02%. The cumulative number of deaths in East China was 43, accounting for 22.75%. Central China, South China, and Southwest China have few fatalities. The injuries of ArRAs are concentrated in Central China and Northeast China, with the two regions accounting for 82.24% of the total number of injuries. Central China had a total of 649 injuries, accounting for 60.04%. A total of 240 people were injured in Northeast China, accounting for 22.20%. However, South China, Southwest China, and North China have few injuries. It is clear that the trends in accidents number, fatalities, and injuries are not in sync. East China has the largest number of accidents, but Northeast China has the most significant number of fatalities, and Central China has the most extensive injuries. Similarly, there are many accidents in Central China but few fatalities. 

#### 3.2.3. Distribution in the Four Economic Zones

In order to further study the relationship between the occurrence of ArRAs and the economic level on a macro scale, the number of ArRAs, casualties, and injuries was statistically analyzed according to the four major economic regions of the country (Northeast, East, Central, and West).

As shown in Figure 7, the Eastern region has the highest frequency of accidents, with 33 (40.24%). They are followed by the Central region, with 21 accidents, accounting for 25.61%. The Western region was the third in the number of accidents with 20. The Northeast region has the lowest number of accidents, but the number of deaths and injuries in the Northeast region is relatively high, with 64.02% of the total fatalities and 22.20% of the injuries. The Central region has the least number of fatalities, but the largest number of injuries, accounting for 60.41% of the total injuries. The West had a relatively low number of fatalities and injuries. Figure 8 clearly shows that the highest number of accidents, fatalities, and injuries are in different economic zones. The Eastern region has a high number of accidents, but the number of deaths and injuries is much lower than in the Northeast.

### 3.3. The Direct Causes of ArRAs

The direct causes of ArRAs are classified and analyzed by equipment failure, material factors, human factors, and external environmental factors. Details are shown in Figure 9.

The human factors mainly include improper operation, misoperation, and illegal operation. There were 17 cases caused by human factors, accounting for 20.73% of the total number of accidents. Material factors include defective welding materials, cracked welds, unqualified insulation materials, broken blind bolts, aging flanges, corroded/aging gaskets, and detached pipe caps, with six cases accounting for 7.32%. Equipment failures mainly contain aging compressors, ruptured tanks, aging pipes, ruptured pipes, loose valves, poorly sealed valves, rusted valves, and leaking valves, which have 55 cases, accounting for 67.07% of the total number of accidents. The external environment mainly manifests in the presence of open flames, abnormal changes in external temperature, and short-circuiting of wiring, with four cases accounting for 4.88% of the total number of accidents.

## 4. Discussion

The main work of this paper is to statistically analyze the characteristics and laws of ArRAs in the past ten years so as to provide a decision basis for the safety management of ammonia-related cold storage. In this section, we analyzed and discussed the results in combination with the current production situation of industrial cold storage, the current status of cold storage safety management, and previous studies. Then, the recommendations for the prevention and control of ArRAs were provided. Finally, some limitations and future improvement directions are given.

### 4.1. ArRAs Time Evolution and Safety Management Policy

As a traditional, natural refrigerant, ammonia is still the first choice in large and medium-sized industrial refrigeration systems [5]. The most crucial consideration in ammonia refrigerant is its safety. Ammonia is toxic, flammable, corrosive, and has an irritating odor [35]. Ammonia-related refrigeration is the focus of safety supervision. There are numerous standards and specifications for cold storage design, construction, operation, and safety management, as shown in Table 5.

The ArRAs number has changed in an “M” pattern over the last decade, during which 2013 and 2016 were the peak years. The number of accidents increased significantly from 2010 to 2013, decreased significantly in 2014 and 2015, increased significantly again in 2016, and decreased yearly since 2017.

The significant decline in accident numbers after 2014 and 2017 was due to the national safety management policies introduced during this period, as shown in Table 6. In 2013, the Safety Committee of the State Council pointed out that special treatment was carried out for ammonia-related refrigeration enterprises to comprehensively improve the intrinsic safety level [8]. The significant reduction in the number of ArRAs in 2014 and 2015 shows that the exceptional treatment work has achieved positive results. In 2016, the state issued successive documents stating that it was necessary to curb the occurrence of severe accidents in the industrial and trade sectors, to grasp the exceptional treatment work in the field of ammonia-related refrigeration, to continue to implement the rectification of major potential accidents involving the use of liquid ammonia in ammonia-related refrigeration enterprises, and to improve accident prevention capabilities [9]. The promotion and implementation of safety policies such as the guidelines for reducing major accidents, the rectification of hidden dangers in ammonia-related refrigeration enterprises, and the identification and control of major risk factors in the industry and trade enterprises have improved the accident prevention capabilities of ammonia-related refrigeration enterprises, resulting in a significant decrease in the number of accidents.

Microscopically, ArRAs have noticeable aggregation effects in the monthly distribution that the accidents mainly concentrated from June to September. Bing Wang et al. [36] and Zimmerman, Laura I et al. [37] suggest that high temperatures are one of the leading causes of accidents involving hazardous chemicals and that accidents frequently occur during the hot season (June to September) in China. The findings of Bing Wang et al. seem to apply equally to ammonia-related refrigeration accidents. Most areas have a high-temperature period from June to September, and the climate is relatively dry. The boiling point of ammonia is −33.34 °C, which is easy to vaporize, leak, and diffuse in an environment with high temperature and low humidity. At the same time, the high ambient temperature makes it more likely to generate an ignition source, and the potential for secondary accidents after leakage is tremendous. In addition, high temperatures influence employees’ psychology and physiology, which significantly impact their behavior.

### 4.2. ArRAs Spatial Distribution and Cold Storage Distribution

The construction and development of cold storage have become the top priority of the national cold chain logistics base [38]. Through 2015–2020, China’s cold storage capacity has had an overall growth rate of more than 10%, including the capacity of 70.8 million tons in 2020, an increase of 17% compared to 2019. Under the influence of the COVID-19 epidemic, governments at all levels realized the importance of cold chain logistics to ensure basic livelihood work. Therefore, promoting cold chain logistics infra-structure layout has become the development of many local government points. According to the cold storage distribution data published by the Research Institute of the Cold Chain Logistics Professional Committee of the China Federation of Logistics and Procurement, the capacity of cold storage in East China ranks first in China, accounting for 36.2%, followed by Central China and North China, accounting for 14.3% and 12.1% respectively. In addition, South China accounted for 10.4% of cold storage capacity, Southwest China accounted for 9.7%, and Northeast China accounted for 8.2%, as shown in Figure 10.

Accidents mainly occurred in East China and Central China from geographical and administrative divisions. From the perspective of economic divisions, ArRAs mainly happened in the Eastern and Central regions. While in terms of the micro-spatial distribution, Zhejiang, Shandong, Sichuan, and Hubei are provinces with high-incidence provinces of ArRAs. East China and Central China are prone to ArRAs, which is consistent with the statistical results of other scholars [33]. Ammonia is a common refrigerant in cold storage. East and Central China have many accidents due to their excellent cold chain infrastructure and adequate cold storage resources.

### 4.3. Variety Trends between the Number of Accidents and Casualties

We find it interesting that the number of accidents is not directly proportional to the number of injuries and fatalities in accidents, whether temporal or spatial distribution. Generally, the higher the number of accidents, the higher the casualties [39,40]. However, as shown in Figure 11, the variety trend in the number of accidents and accident casualties is inconsistent. Figure 11a describes the annual evolution trend of ArRAs number, injuries, and fatalities. Since 2017, the number of ArRAs has decreased, but the number of accident fatalities and injuries has shown an upward trend. This inconsistency is also evident in the monthly distribution, where the months with the highest number of accidents have far fewer casualties than the months with a low number of accidents, as shown in Figure 11b. Figure 11c,d further shows the distribution of the proportion of accidents, fatalities, and injuries in seven administrative geographical regions and four economic zones. Similarly, this asynchronous variation can be seen in the spatial distribution. Eastern China has the highest number of accidents but has few fatalities and injuries. Conversely, the Northeast has few accidents while having the highest fatalities.

The high uncertainty and variability in the consequences of ArRAs may explain this phenomenon. Therefore, the casualty consequences of individual accidents can significantly affect the overall distribution and trends of the three variables. In other words, several major accidents in 2013 may be a fundamental reason for the unsynchronized variety in the three numerical variables. In preventing and controlling accidents in ammonia-related refrigeration, particular attention should be paid to the case-by-case effects and uncertainties of individual accidents. While controlling the total number of accidents, efforts should still be made to prevent and control the occurrence of serious accidents.

### 4.4. Cause Analysis of ArRAs

Human factors and equipment failures are the leading causes of ArRAs, which is consistent with the research results of Liu Hui and Zhang Hu [23,33]. There are 62 accidents caused by these two types of reasons, accounting for 87.80% of the total number of accidents. Both the unsafe behavior and material conditions can attribute to management deficiencies. The unsound safety management system and operating procedures, chaotic site safety management, unlicensed operators, and unimplemented inspection and maintenance operation system are the indirect cause of the accident. The recurrence of ArRAs shows that ammonia refrigeration-related enterprises still have problems of “low level of essential safety, low quality of personnel, and lack of safety risk control capabilities.” Enterprises should further improve and implement safety regulations. For example, operators’ training and education should be strengthened to minimize misoperation and irregularities. Enhance the inspection and maintenance of equipment to improve equipment reliability and ensure safe production.

Furthermore, although the immediate causes of ArRAs are mainly equipment failures and human irregularities, it is also evident that the causes of accidents are not usually singular. In other words, accidents occur due to a combination of causes. The accident causes need to be refined in subsequent studies to clarify further the influence of each factor on the occurrence of accidents so that we can precisely prevent and control ArRAs.

### 4.5. Recommendations for ArRAs Prevention

After the Kigali Amendment to the Montreal Protocol came into effect, many countries are exploring new refrigeration alternatives to reduce the damage to the ozone layer and mitigate global warming [38,41]. However, due to technical and cost constraints, the use of ammonia as a refrigerant is still mainstream in industrial refrigeration, especially in large and medium-sized cold storage. The key to ammonia refrigerated cold storage is safety because ammonia is corrosive, toxic, and has the risk of combustion and explosion when mixed with air [6,35]. Hence, recommendations for the prevention and control of ArRAs are proposed in conjunction with the above research results.

Strengthen the safety education and training for employees. New workers engaged in the use of liquid ammonia positions must not only receive three levels of safety education but also undergo job training to learn job safety operating procedures, production principles, and critical points of safety production (such as the methods of emergency treatment before an accident, protection and rescue knowledge).

Intensify the maintenance and inspection of equipment. Ammonia-related refrigeration enterprises should attach importance to the regular inspection of special equipment and safety accessories as well as the routine maintenance inspection of equipment. At the same time, enterprises can develop an on-site tour inspection system to identify, report, and resolve problems in time, including inspection time, route, and critical parts.

Improve emergency rescue capabilities. Ammonia-related refrigeration enterprises should formulate feasible special and on-site disposal plans for liquid ammonia leakage, poisoning, and explosion, provide emergency rescue personnel and necessary emergency rescue equipment, and organize regular drills.

Enhance the ability of external safety supervision. Further, improve the work safety supervision and management system of ammonia-related refrigeration enterprises. Vigorously implement the “Internet + supervision” and “law enforcement + experts” model to improve the efficiency and effectiveness of safety supervision.

### 4.6. Limitations and Future Research

There are also some shortcomings. The first one is the possible bias of accident data. The lag in the publication of web-based information and the weak event/accident impact may lead to individual accidents/incidents not being identified and counted. The second is the imprecise and scratchy analysis of the temporal distribution characteristics of ArRAs. Based on the completeness and meticulousness of accident information, this study can only explore the annual and monthly distribution characteristics and fails to consider the attributes during different periods of the day. Subsequent studies can further refine the accident pattern at different times of the day to propose more targeted accident prevention countermeasures. At the same time, future research can adopt scenario construction, accident causation analysis, data mining, and other techniques to analyze and explore the relationships between accident causative factors and accidents.

## 5. Conclusions

Ammonia is still the dominant refrigerant in industrial refrigeration. For the safety and sustainable development of ammonia refrigerated cold storage, it is indispensable to study the ArRAs. The present study analyzed the characteristics and regularities of ArRAs in China from 2010 to 2020. This work draws valuable conclusions to inform safety decision-making by ammonia-related refrigeration enterprises and safety regulators.

82 ArRAs from 2010 to 2020 resulted in 189 deaths and 1081 injuries. ArRAs have specific spatial and temporal distribution characteristics. The annual distribution shows a bimodal “M” pattern in the time dimension, with 2013 and 2016 being the peak period of accidents. The monthly allocation shows a clustering effect, and the period from June to September is prone to accidents. In the spatial dimension, accidents mainly occur in East China and Central China, which have abundant cold storage resources. Enterprises and regulators should focus on strengthening the prevention and control of ArRAs in critical regions, provinces and months, and strictly implement the primary responsibility of enterprises for work safety and the responsibility of the government for safety supervision.

The accident number and casualties are out of sync because of the variability and uncertainty of ArRAs’ consequences. Ammonia-related refrigeration enterprises and safety regulators should still focus on preventing and controlling the occurrence of severe accidents while controlling the total number of accidents. The macro safety policy environment is conducive to the safety work of ammonia-related refrigeration enterprises. With the implementation of special treatment in ammonia-related refrigeration policies and the containment of major accidents in the industrial and trade industry, the overall safety work situation of ammonia-related refrigeration enterprises has improved but still needs to strengthen personnel education and training, equipment maintenance and inspection, emergency rescue, and safety supervision to improve the ability of accident prevention and control.

## Figures and Tables

**Figure 1 ijerph-19-08230-f001:**
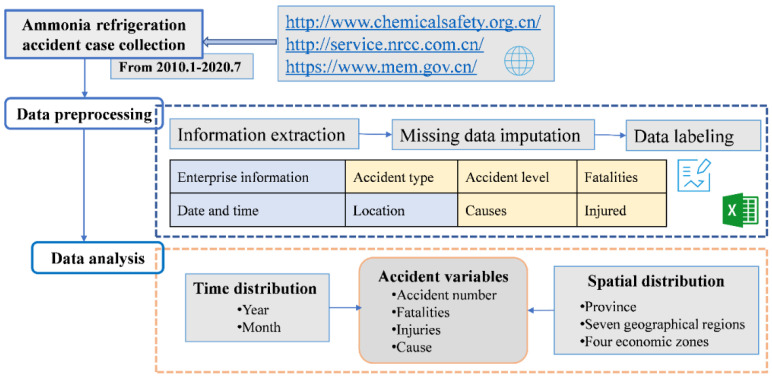
Research flowchart.

**Figure 2 ijerph-19-08230-f002:**
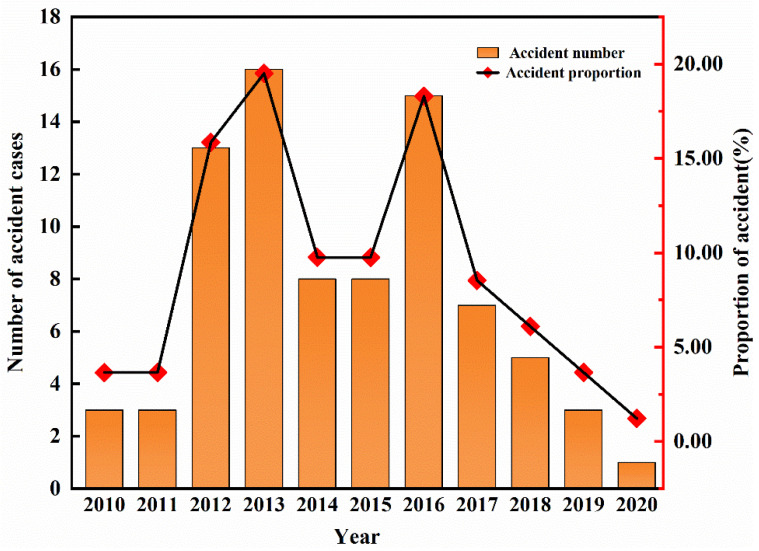
Accident number and proportion by year.

**Figure 3 ijerph-19-08230-f003:**
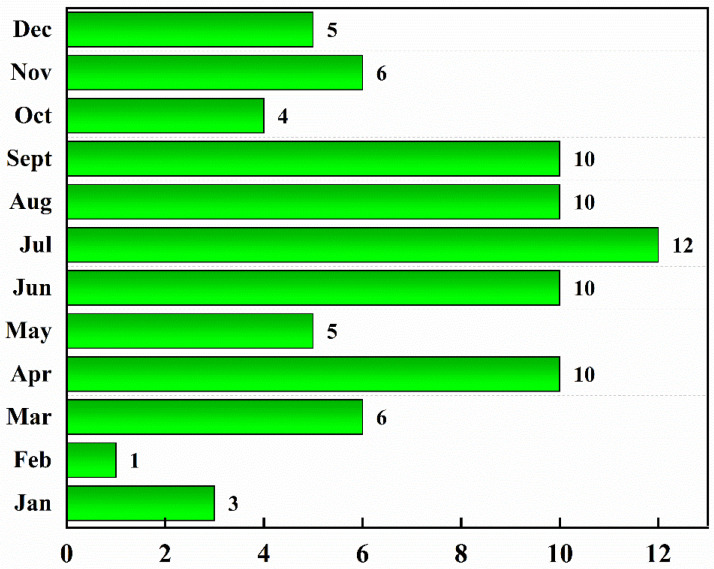
Monthly evolution of ArRAs.

**Figure 4 ijerph-19-08230-f004:**
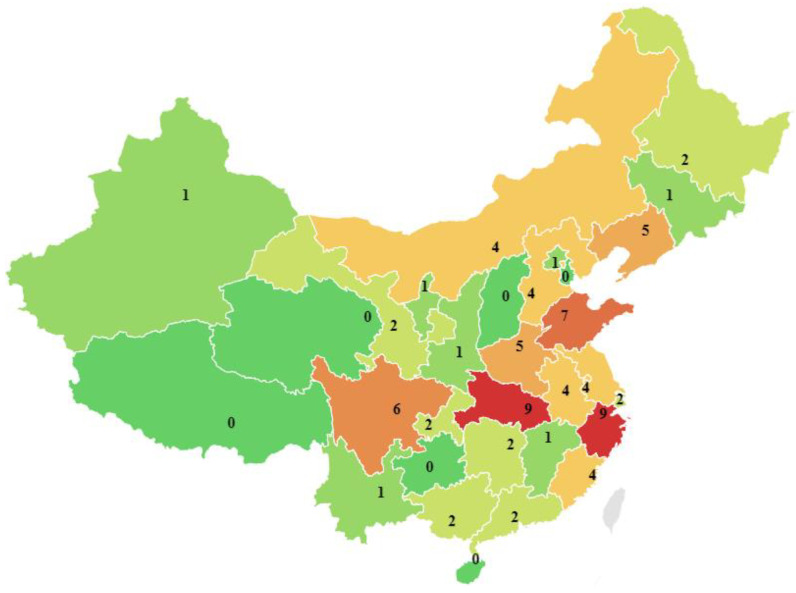
Distribution of ArRAs by 31 provinces (excluding Hong Kong, Macau, and Taiwan).

**Figure 5 ijerph-19-08230-f005:**
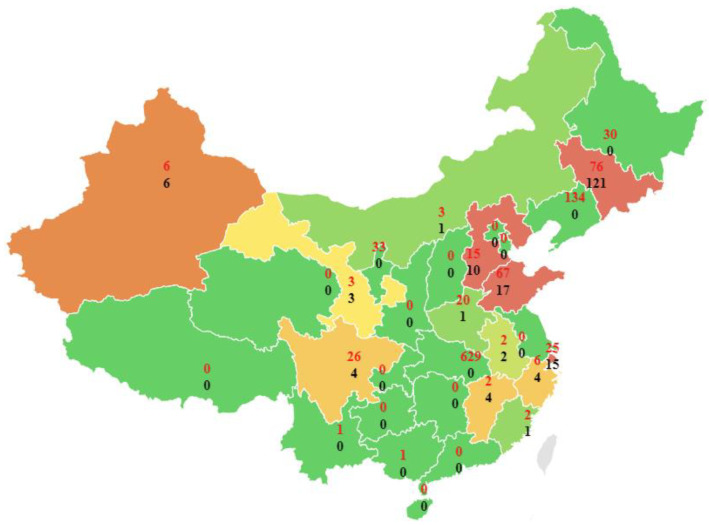
Fatalities and injuries distribution of ArRAs by 31provinces (excluding Hong Kong, Macau, and Taiwan).

**Figure 6 ijerph-19-08230-f006:**
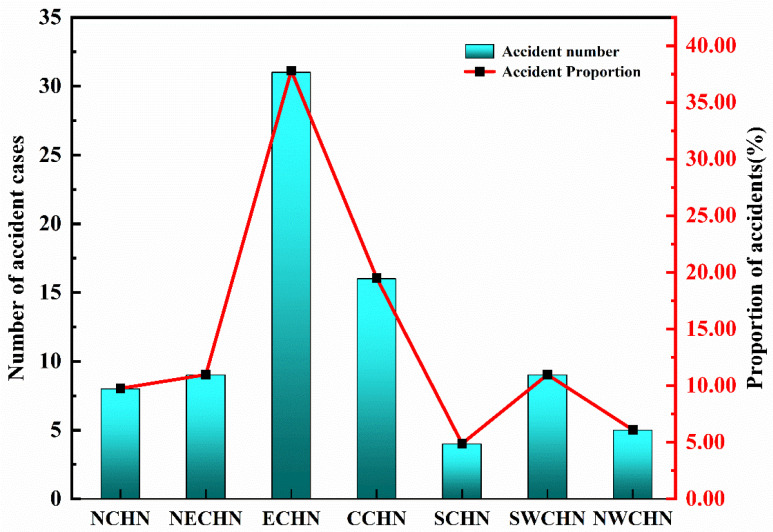
Accident number and proportion by seven administrative divisions.

**Figure 7 ijerph-19-08230-f007:**
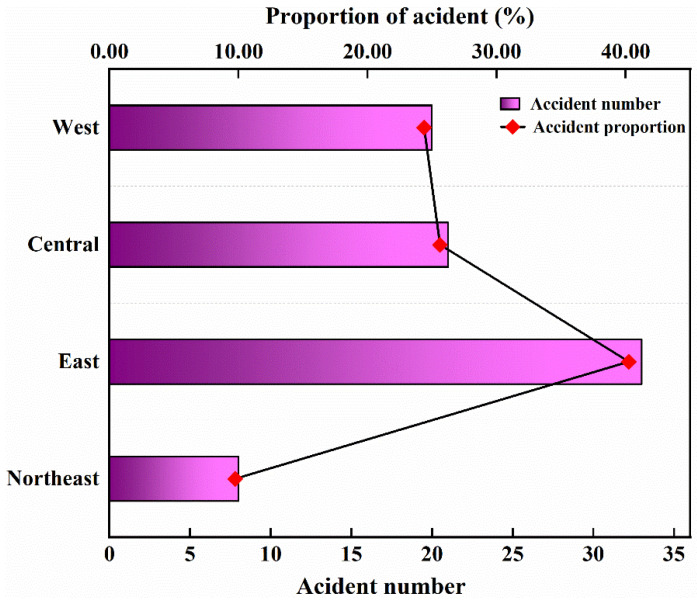
Accident number and proportion by four economic zones.

**Figure 8 ijerph-19-08230-f008:**
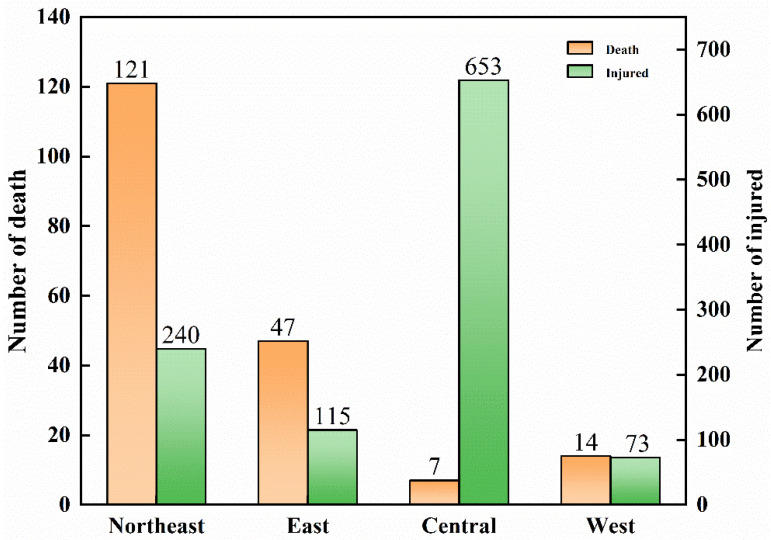
Number of deaths and injured in four economic zones.

**Figure 9 ijerph-19-08230-f009:**
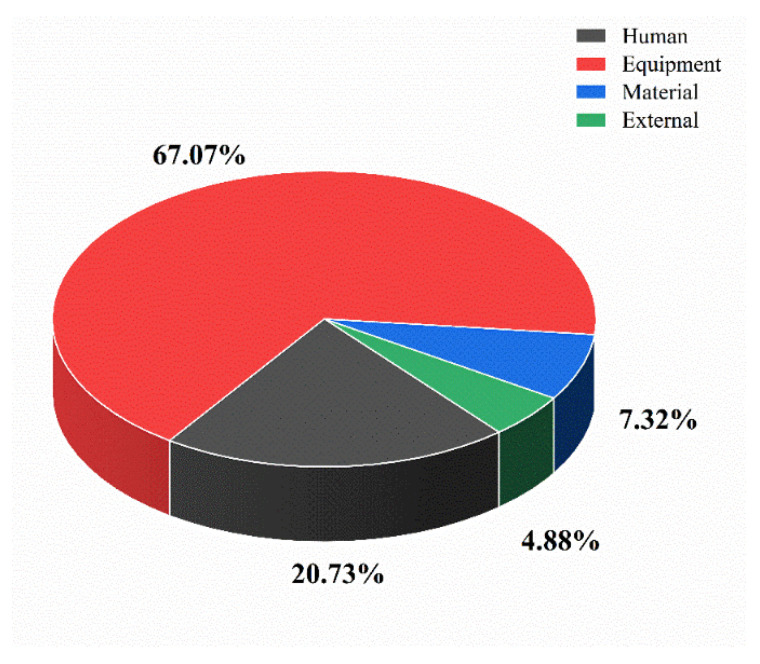
Accident causes statics.

**Figure 10 ijerph-19-08230-f010:**
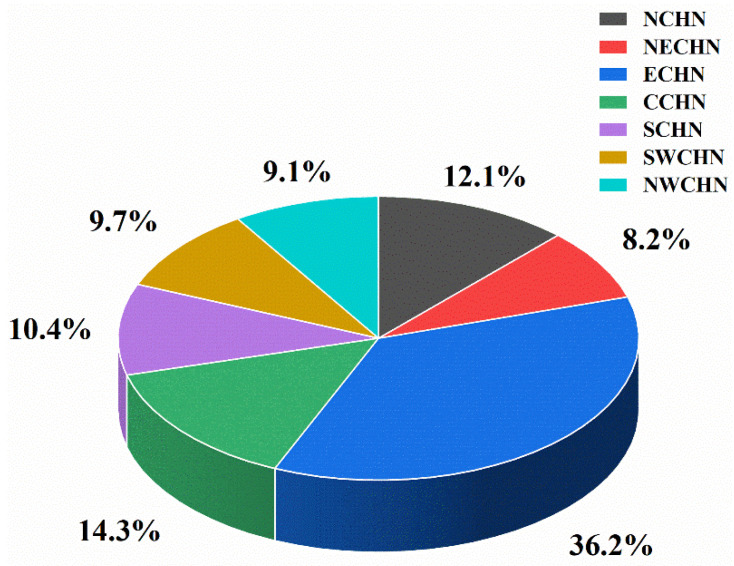
Distribution of cold storage in China.

**Figure 11 ijerph-19-08230-f011:**
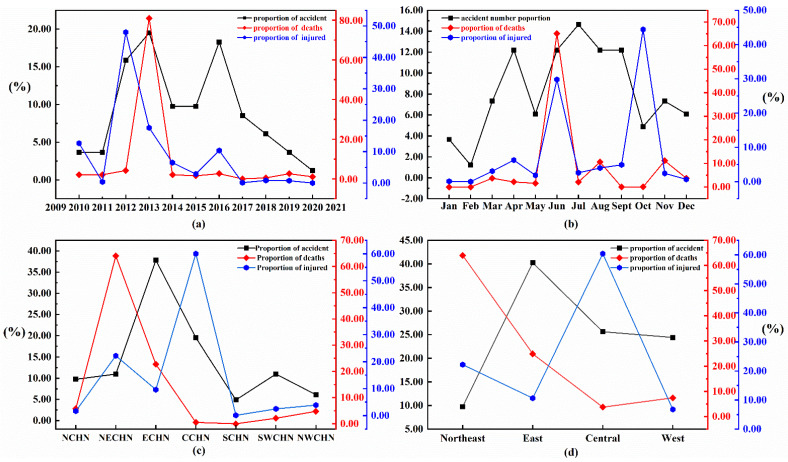
Variety trends between the number of accidents and casualties: (**a**) Variety trend by year; (**b**) Variety trend by month; (**c**) Variety trend by seven geographic regions; (**d**) Variety trend by four economic zones.

**Table 1 ijerph-19-08230-t001:** Number of Accident cases, fatalities, and injuries by year.

Variable	2010	2011	2012	2013	2014	2015	2016	2017	2018	2019	2020	Total
Case	3	3	13	16	8	8	15	7	5	3	1	82
Deaths	4	4	8	153	4	3	5	0	1	5	2	189
Injured	137	4	519	190	70	31	112	1	9	8	0	1081

**Table 2 ijerph-19-08230-t002:** Typical cases of ArRAs.

Time	Name and Addressess	Fatalities	Injuries
28 August 2011	Hebei Zhangjiakou Jia Green Agricultural Products Company (Zhangjiakou, China)	4	4
23 October 2012	Hubei Honghu City Deyan Fisheries Company (Honghu, China)	0	479
21 April 2013	Sichuan Meishan Renshou County Jinfeng Food Factory (Meishan, China)	4	22
3 June 2013	Jilin Dehui Baoyuanfeng Poultry Industry Co. (Dehui, China)	121	76
31 August 2013.	Shanghai Baoshan Weng Refrigeration Industry Co. (Shanghai, China)	15	25
28 November 2013	Shandong Yushan Hehe Food Co. (Yushan, China)	7	6

**Table 3 ijerph-19-08230-t003:** Number of Accident cases, fatalities, and injuries by month.

Month	Accident Case		Fatalities		Injured	
	Number	Proportion	Number	Proportion	Number	Proportion
January	3	3.66%	0	0.00%	1	0.09%
February	1	1.22%	0	0.00%	0	0.00%
March	6	7.32%	7	3.70%	33	3.05%
April	10	12.20%	4	2.12%	68	6.29%
May	5	6.10%	3	1.59%	20	1.85%
June	10	12.20%	123	65.08%	322	29.79%
July	12	14.63%	4	2.12%	28	2.59%
August	10	12.20%	20	10.58%	43	3.98%
September	10	12.20%	0	0.00%	53	4.90%
October	4	4.88%	0	0.00%	480	44.40%
November	6	7.32%	21	11.11%	26	2.41%
December	5	6.10%	7	3.70%	7	0.65%

**Table 4 ijerph-19-08230-t004:** Number of accidents, fatalities, and injuries by seven administrative geographical regions.

Region	Accident Case		Fatalities		Injured	
	Number	Proportion	Number	Proportion	Number	Proportion
North China	8	9.76%	11	5.82%	18	1.67%
Northeast China	9	10.98%	121	64.02%	240	22.20%
East China	31	37.80%	43	22.75%	104	9.62%
Central China	16	19.51%	1	0.53%	649	60.04%
South China	4	4.88%	0	0.00%	1	0.09%
Southwest China	9	10.98%	4	2.12%	27	2.50%
Northwest China	5	6.10%	9	4.76%	42	3.89%

**Table 5 ijerph-19-08230-t005:** Main standards and specifications in ammonia-related refrigeration field.

Year	Laws, Standards, and Specifications
2021	Work Safety Law of the PRC
2014	Special Equipment Safety Law of PRC
2013	Regulations on the Safe Management of Hazardous Chemicals in China
2012	Regulation on Safety Operation Licenses for Hazardous Chemicals Enterprises
2009	Regulations on safety supervision of special equipment
2011	Interim Provisions on the supervision and administration of major hazard installations of hazardous chemicals
2010	Management regulations for safety technical training and assessment of special operation personnel
2010	GB 50072 Code for design of cold store
2011	GB 28009 Safety code for cold store
2018	GB 18218 Identification of Major Hazard Installations
2014	GB 50016 Code for Fire Protection Design of Building
2020	GB/T 20801 Pressure piping code—Industrial piping
2014	GB 50351 Code for design of fire-dike in storage tank farm
2018	AQ 7015 The safety specification of ammonia refrigeration user
2011	SBJ 12 Code for installation and acceptance specification of ammonia refrigeration system
2019	DB44/T 2161 Cold storage safety management specification

**Table 6 ijerph-19-08230-t006:** Safety management policy related to ammonia refrigeration.

Number	Name	Time
1	Notice of the Safety Committee of the State Council on In-depth Implementation of Special Treatment of Liquid Ammonia Use in Ammonia-related Refrigeration Enterprises (Safety Committee [2013] No. 6)	24 September 2013
2	Notice of the State Administration of Safety Supervision on the Issuance of Opinions on Curbing Serious Accidents in the Industry and Trade Sector (General Administration of Safety Supervision IV [2016] No. 68)	28 June 2016
3	Notice of the Office of the Safety Committee of the State Council for urging ammonia-related refrigeration enterprises to rectify major hidden accidents and strengthen safety supervision (Safety Committee Office Letter [2016] No. 3)	25 February 2016
4	Notice of the State Administration of Safety Supervision on the Issuance of the Action Plan to Carry out Identification and Control of Larger Risk Factors in Industrial and Trade Enterprises to Enhance the Ability to Prevent Accidents (General Administration of Safety Supervision IV [2016] No. 31)	1 April 2016

## Data Availability

The date can be obtained from the corresponding author upon reasonable request.

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
