# Peer review of "Study on the Regularity of Ammonia-Related Refrigeration Accidents in China from 2010 to 2020"

_ijerph, 2022, doi:10.3390/ijerph19148230_

Round 1
Reviewer 1 Report
1. The reviewer wonders what is the originality of this paper? Only the statistical method used in this paper is original? But this method is not proposed by the authors.
2. References cited in section 3.1,3.2 and 4.3 are not found. The authors should double check this issue before submission.
Reviewer 2 Report
The topic of the paper, and its research questions, are highly relevant and of great practical interest
The paper demonstrates knowledge of comparable methods presented in the literature.
The case study is an important part of the paper. The proposed methodology is very interestig.
Some issues that may enhance the contribution of this work:
- the authors should better explain the figure 1;
- the authors should better explain the questionnarie structure and the reasons (it is a highly relevant part of your paper)
- the authors should correct some grammatical errors
Please improve english and references, e.g.:
-Systematic Human Reliability Analysis (SHRA): A New Approach to Evaluate Human Error Probability (HEP) in a Nuclear Plant Bona, G.D., Falcone, D., Forcina, A., Silvestri, L.International Journal of Mathematical, Engineering and Management Sciences,
-Quality Checks Logit Human Reliability (LHR): A New Model to Evaluate Human Error Probability (HEP)
Di Bona, G., Falcone, D., Forcina, A., De Carlo, F., Silvestri, L. ,
On risk-based maintenance: A comprehensive review of three approaches to track the impact of consequence modelling for predicting maintenance actions
Leoni, L., De Carlo, F., Paltrinieri, N., Sgarbossa, F., BahooToroody, A. ,Proposal of a multidimensional risk assessment methodolgy to assess ageing workforce in a manufacturing industry: A pilot case study
De Felice, F., Longo, F., Padovano, A., Falcone, D., Baffo, I. ,Author Response
Point 1:the authors should better explain the figure 1;
Response 1: Figure 1 shows the research idea of this paper, which depicts the data source, data processing, and data analysis. In response to the reviewers' comments, we have revised Section 2 further to strengthen the connection between figure 1 and the text, thus reducing possible confusion and doubts of the readers.
Point 2: the authors should better explain the questionnarie structure and the reasons (it is a highly relevant part of your paper)
Response 2: Many thanks to the reviewer for your suggestions. We focused on supplementing and improving the background in the introduction section so that the reader can understand more clearly why we conducted this study and proposed this research question. Also, we have revised the abstract, discussion, and conclusion paragraph. We have also added a description of the paper structure.
Point 3: the authors should correct some grammatical errors
Response 3: We have thoroughly checked the paper and corrected the grammatical errors and misrepresentations. We also invited experienced Dr. Guo to check the revised manuscript before submitting it.
Point 4: Please improve english and references
Response 4: We checked and improved the references according to the journal's guidelines and the reviewers' comments. Specifically, we added some essential and classic references and revised the reference format to meet the journal's requirements.
Reviewer 3 Report
This is one of the finest papers I have read lately. It is well worth publishing, provided minor flaws are corrected, the details of which are in the attached report.

Author Response
Point 1: On quite a number of occasions, the reference in the text to a Figure or to a Table has led to the mention "Error! Source not detected”: lines 130, 143, 151, 154, 175, 182, 238;
Response 1: We have checked the cross-reference of figures and tables in the manuscript and revised the reference in lines 130, 143, 151, 154, 175, 182, 238, and 337.
Point 2: The four graphs in Figure 11 are a bit small and therefore a somewhat difficult to read;
Response 2: We redrew Figure 11, focusing on the figures' font and spacing to provide a more straightforward reading.
Point 3: The authors should be more assertive in their recommendations; weak recommendations are in direct contradiction with the first sentence of the conclusion (cited above); given that the authors have identified specific regions and periods of the year where accidents tend to occur more, some specific recommendations might be proposed in that respect.
Response 3: Thanks to reviewer for the constructive comments. In the beginning, the research focused on exploring accident characteristics and regularities (a kind of facts and evidence to a certain extent) to provide a reference for relevant enterprises and regulatory authorities. Therefore, we did not put much emphasis on the recommendation piece. In other words, we prefer to emphasize an epistemology of ArRAs rather than a methodology to control ArRAs. Clearly, as the reviewer notes, inadequate and weak suggestions weakened our paper. For this reason, we have reworked the recommendations in section 4. Targeted recommendations for accident prevention and control were presented in conjunction with the research results.
Reviewer 4 Report
The authors submitted an interesting study that analyzed the characteristics and regularities of ammonia-related refrigeration accidents (ArRAs) in China from 2010 to 2020. This work draws valuable conclusions to in-form safety decision-making by ammonia-related refrigeration enterprises and safety regulators. The work has a good structure and has the potential for practical use. The paper is suitable for the MDPI journal "Environmental Research and Public Health". I have a few notes for the authors:
1. Please correct erroneous references to images in the text (see "Error! Reference source not found .." in pdf). Check the entire document.
2. Why has the number of accidents decreased since 2016? Have any safety laws been passed?
3. The authors should discuss more the current safety and health conditions at work in China and how often are inspections made.
4. Ammonia is corrosive and high concentrations of ammonia in the air can cause blindness and lung damage. Many US companies are abandoning the use of ammonia in refrigerators. The authors should discuss when this trend will be on the rise in China as well.
5. What are the practical impacts of implementing a safety management policy over the years (see Table 5)? Please discuss it.
6. The timeline in Figure 2 should have a caption.
7. The abbreviations on the horizontal axis in Figures 6 and 10 should be explained!
8. In discussion, authors should discuss the results and how they can be interpreted from the perspective of previous studies and of the working hypotheses. The findings and their implications should be discussed in the broadest context. Future research directions may also be highlighted. The practical use of the proposal should be more emphasized.
I suggest accepting the paper, but my comments should be resolved.
Author Response
Point 1: Please correct erroneous references to images in the text (see "Error! Reference source not found .." in pdf). Check the entire document.
Response 1: We have checked the cross-reference of the figure and table in the manuscript and revised the reference in lines 130, 143, 151, 154, 175, 182, 238, and 337.
Point 2: Why has the number of accidents decreased since 2016? Have any safety laws been passed? What are the practical impacts of implementing a safety management policy over the years (see Table 5)? Please discuss it.
Response 2: As the reviewer stated, the number of accidents declined after 2016. The accident number in 2016 was 15, while in 2017, 2018, 2019, and 2020 it was 7, 5, 3, and 1, respectively. We speculate that the decrease in accidents is related to the safety management policy issued in 2016, discussed in Section 4.1. In 2016, the state announced three safety management requirements: Notice of the State Administration of Safety Supervision on the Issuance of Opinions on Curbing Serious Accidents in the Industry and Trade Sector (General Administration of Safety Supervision IV [2016] No. 68), Notice of the Office of the Safety Committee of the State Council for urging ammonia-related refrigeration enterprises to rectify major hidden accidents and strengthen safety supervision (Safety Committee Office Letter [2016] No. 3), Notice of the State Administration of Safety Supervision on the Issuance of the Action Plan to Carry out Identification and Control of Larger Risk Factors in Industrial and Trade Enterprises to Enhance the Ability to Prevent Accidents (General Administration of Safety Supervision IV [2016] No. 31). Ammonia-related refrigeration enterprises are the focus of industrial and trade safety supervision, so they are also the key to curbing major accidents in the industrial and trade sector. The promotion and implementation of safety policies such as the guidelines for reducing major accidents, the rectification of hidden dangers in ammonia-related refrigeration enterprises, and the identification and control of major risk factors in the industry and trade enterprises have improved the accident prevention capabilities of ammonia-related refrigeration enterprises, resulting in a significant decrease in the number of accidents after 2016.
Point 3: The authors should discuss more the current safety and health conditions at work in China and how often are inspections made.
Response 3: We thank the reviewers for the constructive comments. Indeed, we did not discuss the safety status and inspection in detail. Therefore, we made some changes. In section 4.1, we added content about the current safety situation of ammonia-related refrigeration enterprises and supplemented laws and regulations related to ammonia-related refrigeration safety management. At the same time, combining the research results with the current situation of cold storage safety management and the future development trend of cold storage, suggestions for the prevention and control of ArRAs are put forward.
Point 4: Ammonia is corrosive and high concentrations of ammonia in the air can cause blindness and lung damage. Many US companies are abandoning the use of ammonia in refrigerators. The authors should discuss when this trend will be on the rise in China as well.
Response 4: Ammonia is toxic and corrosive, as the reviewer says. After the Kigali Amendment to the Montreal Protocol came into effect, many countries are researching new refrigeration alternatives to reduce damage to the ozone layer and mitigate global warming. The abandonment of ammonia is only in areas such as commercial refrigeration, air conditioning, and refrigeration of household appliances. In fact, the use of ammonia remains mainstream in industrial refrigeration, especially in large and medium-sized cold storage. We believe that ammonia still has much room for development in the industrial refrigeration field. Mainly for the following considerations.
- The superiority of ammonia as a refrigerant. Ammonia is a natural refrigerant with good thermodynamic properties, such as large refrigeration capacity per unit mass, moderate condensing temperature, small viscosity, and good thermal conductivity. Secondly, ammonia is easy to obtain, inexpensive, almost insoluble in oil, has low flow resistance and is easy to detect in case of leakage. Although ammonia is toxic and can also be subject to combustion and explosion, the potential hazards can be avoided by reducing the amount of ammonia charge, improving equipment reliability, and monitoring and warning technology.
- Good policy guidance. In recent years, the State Council, the Ministry of Commerce, the Ministry of Finance, and the Ministry of Agriculture have issued several cold chain logistics infrastructure-related policies to support the construction of China's cold chain logistics infrastructure. Cold storage ushered in a favorable policy as an important core of the cold chain infrastructure. 2015-2020 China's cold storage capacity has an overall growth rate of more than 10%, including the capacity of 70.8 million tons in 2020, an increase of 17% compared to 2019. In addition, under the influence of the COVID-19 epidemic, cold chain logistics has become an important channel for linking global fresh food. Governments at all levels are aware of the importance of cold chain logistics in ensuring basic livelihood work. Therefore, the development of the cold chain logistics industry and the promotion of cold chain logistics infrastructure layout have become the development points of many local governments.
- The huge demand for cold storage. Compared to the huge demand for fresh markets, the cold storage capacity growth rate still has a large space. Although the total cold storage capacity has been greatly improved, China's per capita cold storage capacity is only 0.13 cubic meters, far below the level of the United States (0.49 cubic meters), Japan (0.32 cubic meters), and South Korea (0.28 cubic meters). In other words, compared with developed economies, China's cold storage capacity is still insufficient, and the market development space is vast.
Point 5: The timeline in Figure 2 should have a caption.
Response 5: Added axis titles to the time axis of Figure 2 and checked the other figures' axes.
Point 6: The abbreviations on the horizontal axis in Figures 6 and 10 should be explained!
Response 6: Complemented the description of the abbreviations on the horizontal axes in Fig. 6 and Fig. 10.
Point 7: In Discussion, authors should discuss the results and how they can be interpreted from the perspective of previous studies and of the working hypotheses. The findings and their implications should be discussed in the broadest context. Future research directions may also be highlighted. The practical use of the proposal should be more emphasized.
Response 7: We have revised the Discussion in section 4 according to the reviewer's comments. We further strengthen the comparison with other scholars' studies in the discussion and provide a more in-depth explanation of the research results. We also further discussed the significance and practical use of our research in the current situation of cold storage development in China. Finally, future research directions are proposed.
Round 2
Reviewer 1 Report
The manuscript can now be accepted.
Reviewer 2 Report
Paper is ok